**Data Availability Statement:** This is a qualitative study with individual and focus group interviews. The full dataset contains personal information

# Exploring factors affecting the facilitation of nursing students to learn paediatric pain management in Rwanda: A descriptive qualitative study

Philomene Uwimana[1]⊙*, Donatilla Mukamana[1]⊙, Yolanda Babenko-Mould[2]⊙, Oluyinka Adejumo[1]⊙

1 School of Nursing and Midwifery, College of Medicine and Health Sciences, University of Rwanda, Kigali, Rwanda, 2 Faculty of Health Sciences, Arthur Labatt Family School of Nursing, Western University, London, Ontario, Canada

⊙ These authors contributed equally to this work.
* philouwim@gmail.com

## Abstract

Nurse educators and nurse preceptors play a fundamental role in facilitating nursing students' acquisition and utilization of professional competencies. Previous studies about key elements for teaching and learning about pain in nursing education programs include students' personal characteristics and previous experiences; educators' knowledge, skills, and beliefs; learners' exposure to leaders in pain education; and curricular pain content and delivery approaches. These studies were mainly carried out in developed countries, with a context of educational and health care systems different from those of developing countries. The current study explores academics', clinical nurse preceptors', and nursing students' perceptions about factors influencing the facilitation of nursing students' competency for paediatric pain management in Rwanda. A qualitative descriptive exploratory design was used in this study that utilized in-depth interviews with six nurse educators and eight nurse preceptors, and focus group discussions with nineteen senior year nursing students. The study setting included five sites: two academic institutions and three clinical settings. Narratives from participants were transcribed verbatim and analysed using thematic analysis. The analysis yielded six themes describing factors that affected the facilitation of students' learning about paediatric pain management. The themes included student motivation, facilitators' attributes, collaboration between academics and clinicians, nurses' limited autonomy for decision-making regarding PPM practices, shortage of human and material resources, and educational qualification. Knowing these factors is essential as it provides an opportunity to design targeted interventions aimed to enhance the capacity of nurse educators and clinical nurse preceptors involved in teaching nursing students about paediatric pain management.

about the participants; therefore, it cannot be shared to the public. Nonetheless, the minimal dataset underlying this study can be found in the paper and its Supporting Information files.

**Funding:** The Training Support Access Model (TSAM/MNCH) for Rwanda (https://tsam.uwo.ca/), a Western University project funded by Global Affairs Canada, financially supported the study data collection related expenses. There was no involvement of funders in designing the study protocol, data collection and analysis, or in writing the manuscript.

**Competing interests:** The authors have declared that no competing interests exist.

## Introduction

Efforts have been made to utilize effective means for pain relief, yet children continue to suffer unnecessarily. While optimal pain relief is a recognised right for children [1], evidence indicates that pain in children remains under-assessed and under-managed [2,3]. Poorly or under-managed paediatric pain can lead to a prolonged hospital stay for the paediatric patient, and the risk of developing chronic and persistent pain [4,5], as well as other implications for the psychosocial, spiritual, and financial state of the entire family, associated with the child's delay in recovery [6,7]. All these concerns highlight the importance of the optimal management of paediatric pain.

Nurses play a critical role in the effective management of pain in children because they have more contact with paediatric patients than other healthcare professionals. Pain is considered as a 'nursing-sensitive indicator'[8, p.215]; hence nurses ought to perform comprehensive pain assessment and enact effective pain relief to help prevent unnecessary complications in paediatric patients. Optimal nursing management of pain in children requires nurses to be knowledgeable about pain mechanisms, the epidemiology of pain, barriers to effective pain relief, commonly encountered pain conditions, and appropriate methods to assess and alleviate the pain [9]. In this regard, nurse educators and nurse preceptors play a fundamental role in facilitating nursing students' acquisition and utilization of pain management competencies as the next generation of nurses. The role of nurse academicians and preceptors to collaboratively prepare nursing students achieving their professional goals and to develop work-ready nurses has been emphasized by other researchers [10,11].

Undergraduate nursing education in Rwanda is provided through a three- or four-year educational program for a diploma or baccalaureate degree, respectively. Apart from the theoretical knowledge acquired by nursing students through classroom sessions and practice using simulation-based education, educators work with nurse preceptors and nursing students in hospitals to ensure students have opportunities to engage in clinical learning experiences during their practice-based rotations [12]. Academics and practice partners are involved in the process of facilitating students' development of new knowledge which becomes reinforced during students' clinical practice [13]. It is during those educational sessions that students learn nursing competencies including principles of paediatric pain management (PPM). Studies external to Rwanda that examined pain management education indicated a variety of factors that affect the effective facilitation of nursing students to learn PPM. A qualitative study conducted in the United States indicated that knowledge, attitudes, and previous experiences of nurse educators were relevant and influential in the provision of pain education [14]. A review of the literature about pain education revealed that students' characteristics and previous experiences; educators' knowledge, skills and beliefs; exposure to leaders in pain education; pain content and delivery approaches, and pain competencies were key elements for teaching and learning about pain [15].

In a recent qualitative study conducted in Rwanda, nursing students reported concerns related to their learning about PPM, which included unclear teaching plans, ineffective role modelling for students, perceived paediatric pain negligence, and facilitators' knowledge and skill deficits. This study in Rwanda identified challenges that hamper the way nursing students learn to manage pain in children [16]. The findings further revealed an urgent need to explore the factors influencing nurse educators' and preceptors' facilitation of nursing students' competency acquisition regarding PPM [16]. It is believed that a clear understanding of those factors could highlight appropriate interventions to improve PPM education for nursing students. Therefore, the current study built on the recommendations from the previous study

and explored factors affecting the facilitation of nursing students for learning PPM as perceived by academics, clinical nurse preceptors, and nursing students in Rwanda

## Materials and methods

### Study design and settings

The study utilized a qualitative descriptive exploratory design [17]. Descriptive studies strive to gain an in-depth understanding while allowing researchers to obtain a rich description from participants' perspectives of a phenomenon under investigation [17–19]. The use of the qualitative descriptive approach is considered by previous researchers as important and relevant for studies seeking descriptions of experiences and perceptions of participants regarding a phenomenon which is poorly understood [18–21]. Little is known about factors influencing the facilitation of nursing students to learn PPM in the Rwandan context, thus a qualitative descriptive approach was deemed appropriate for the current study to explore and describe perceptions of nurse educators, nurse preceptors, and nursing students about the topic. Also, Neergaard stated that qualitative descriptive approach is relevant for research using mixed methods [17]. The current study is part of a multiphase study that used mixed methods. Qualitative descriptive studies usually use semi-structured interviews with open-end questions either with individuals or with focus groups but also can be individual and focus group interviews. Semi-structured in-depth interviews allow a deep exploration of participants' perspectives about a particular topic, while assessing their individual understanding, values, beliefs, experiences, and perceptions [22,23]. In this study, in-depth interviews (IDIs) with nurse educators and nurse preceptors were conducted to get insightful views about the factors that affect the facilitation they provide to nursing students learning PPM. Whereas the researchers opted for focus group discussions (FGDs) with senior year nursing students, seeking their diverse perceptions of what impede on their learning about PPM.

The study setting included five sites of which two were academic institutions and three were clinical settings. To reach a variety of study sites, these were chosen purposively taking into consideration whether they are public or private institutions, if they are located in a rural or urban area, and for teaching institutions, if an advanced diploma and/or bachelor degree program of nursing is offered. One of the two selected academic institutions is private, located in a rural area, and offers a diploma nursing program, while the other is a public institution offering both diploma and baccalaureate nursing programs and located in an urban area. The three clinical settings involved one teaching hospital, one referral hospital, and one district hospital. These healthcare institutions also serve as clinical practice settings for nursing students during their clinical placement rotations. The teaching hospital and the referral hospital are public and situated in the urban area, whereas the district hospital is a subsidised faith-based institution located in a rural area.

### Selection of participants

Study participants were purposively selected based on criterion sampling for being a nurse educator; having facilitated nursing students to learn PPM in the classroom, in simulation, and/or in the clinical setting; and consenting to participate in the study. Nurse preceptors were also invited to participate in this study given their role of facilitating nursing students' learning practices during their clinical rotations. Consideration was given to the length of time having worked in the paediatric unit and having facilitated learning for nursing students in the past six months. The recruitment of nurse educators was done via email sent to them by one of the researchers and those who responded showing their interest in participating in the study were approached for further arrangement regarding their preferred location and time for the

interview. For nurse preceptors, study members coordinated with matrons (in charge nurses/ unit managers) of paediatric units at the hospitals to discuss the objective of the prospective study during nursing staff meetings, after which one of the research team members addressed any study questions that preceptors had and then follow-up with preceptors interested in participating in the study to schedule interviews.

After interviewing six nurse educators and eight preceptors, the researchers could not collect any new information responding to the objective of the study, indicating that data saturation was achieved [21,24,25]. Neergaard [17] recommended a minimum variation sampling to get a broad insight into a subject of interest. In this study, the variation of participants was observed in their demographic characteristics related to the gender, age, occupation, educational qualification, employment setting, having received a training on PPM after the nursing basic education, and the experience of facilitating students learning PPM.

Third-year students in the diploma nursing program and fourth-year students in the bachelor's nursing program were also invited to participate in the study since they are the key stakeholders in the learning process regarding PPM. These were senior year nursing students who had completed both the theory and clinical parts of paediatric nursing course and could identify factors that promoted or hindered their learning about PPM., as they are expected to possess the required PPM competency for future nursing practice. Students who participated in this study were recruited through the head of nursing department at the participating Schools with whom the research team liaised to acquire permission to contact students. Nineteen nursing students (9 from the diploma program and 10 from the bachelor's program) consented to participate in the study and constituted the final sample size for the FGDs. The sample size of the FGDs was deemed sufficient in this study considering what was stated in the literature [26]. Student participants consisted of thirteen males and six females with an average age of 24 years (range of 22 to 28 years); and were recruited from a private academic institution (n = 9), and a public academic institution (n = 10).

For the confidentiality of participants, they are identified as NE1, 2, 3,... for nurse educators; P1, 2, 3.... for preceptors, and S1, 2, 3... for students.

## Data collection procedures

The research team used semi-structured interview guides to support relevant information was obtained from participants. The interview guides were developed in English, then translated to Kinyarwanda (S1 and S2 Files). This was done with the assistance of a professional translator and checked by the research team for the authenticity of the meaning and content of the items vis-à-vis their originality. The IDIs and FGDs were conducted either in English or Kinyarwanda depending on the choice of the participant(s). With the semi-structured interview guides, participants were given an opportunity to openly share their experiences and perceptions regarding the factors influencing the facilitation of nursing students to learn PPM. Among the questions included in the IDIs interview guide, participants were asked to describe how capable nurse educators and preceptors in their settings were in facilitating students learning PPM and what were the challenges or concerns they had relating to the facilitation of students' learning PPM (S1 File). The interviewer actively listened to participants' responses in order to clarify or further explore participants' responses using probing questions whenever it was necessary.

In-depth interviews (IDIs) with nurse educators and preceptors were held at a place that was convenient to the participants and where they felt that their privacy was preserved, either at school in the office of the nurse educator or the office of the unit manager at the hospital. Participants were interviewed in person by one member of the research team (PU). The IDIs ranged in length from 26 to 60 minutes with a mean of 40 minutes.

Focus group discussions (FGDs) were conducted with nursing students to inquire about their perceptions regarding the effectiveness of facilitation they received from nurse educators and preceptors to learn PPM, and to describe any difficulties or concerns encountered in their learning about PPM. For instance, a probing question in the FGDs was "What would you want to be changed, and in what ways, about students' facilitation to learn PPM by nurse educators and preceptors?" (S2 File).

We considered two FGDs sufficient to enable data saturation, recognizing the literature notes that two to three focus groups can lead to discovering more than 80% of themes [27]. The FGDs were organised at the nursing school and clinical placement setting, and were held in a quiet private room where only participants and the researcher who was overseeing the FGDs were present. The length of each FGD ranged between 60 to 70 minutes.

Before ending of each interview, the primary investigator who conducted the interviews and facilitated the FGDs at the different times provided to the participants the opportunity to discuss any additional perspectives related to the facilitation of students' competency acquisition for PPM that were not shared during IDI or FGD. Moreover, toward the end of each IDI or FGD, the researcher did a recap of the main points discussed during the session for clarifications and corrections where it was necessary.

All participants consented to audio-recording of their responses, while confidentiality was preserved with the use of number codes and pseudonyms to represent the participants. The data collection process took place over three months, from November 2018 to January 2019.

## Ethical considerations

The study was completed after considering ethical implications, complying with the provisions of the Declaration of Helsinki [28]. This study was approved by the Institutional Review Board of the College of Medicine and Health Sciences at the University of Rwanda (No:341/CMHS IRB/2018). Also, the management of the hospitals and nursing schools where the study was conducted provided permission to collect data, and written informed consent was obtained from the study participants.

## Data analysis

Verbatim transcribing of the audio-recorded data was completed and then transcripts were analysed using thematic analysis steps as described by [29]. The thematic analysis process helps to summarise data sets and provides structured guidance to handle data and produce a clear and organised report [30]. To verify the completeness and accuracy of the transcribed data, the transcripts were read several times while listening to the audio-recorded data. NVivo version 12 Pro was used as a data management aid [31]. The research team read and re-read the transcripts to familiarise themselves with the content before engaging in the coding process. At the beginning, we analysed the same transcript, each research team member independently, and then discussed the emerging themes with the purpose of building consensus through the coding process to thematic identification.

The coding process was interactive using the line-by-line technique to create initial codes. Then the initial codes were combined into larger meaningful segment codes from which the themes emerged. The use of inductive coding approach allowed the development of unanticipated themes from the raw data [32]. The analytic technique was reinforced by the researchers' interest in nurse educators', preceptors', and nursing students' perceptions of factors influencing the facilitation of competency acquisition for PPM by nursing students. During the data analysis process, the research team held several discussions about the generated codes, ensuring that the coding was completed and any divergence was resolved through consensus

building. Data analysis was completed when no more new themes or sub-themes emerged from participants' data.

## Validity and rigour

For the validity and rigour of the present study, the researchers adhered to the principles of trustworthiness that include credibility, confirmability, dependability, and transferability [33,34]. To ensure credibility, only participants who met the study inclusion criteria were recruited. The anonymity and confidentiality of participants were preserved during the data collection process and data storage. Also, checking the transcripts for accuracy by research team members as well as checking of the generated themes by experienced qualitative researchers were utilized to ensure credibility and confirmability of the findings. Furthermore, a detailed description of the study settings, participants' inclusion criteria, sampling and data collection procedure, and data analysis method was shared among team members, enhancing the dependability and transferability of the study findings.

## Results

The nurse participants consisted of eight females and six males; their ages ranged from 28 to 46 years with more than three-quarters being over 31 years of age. Student participants were thirteen males and six females with an average age of 24 years (range of 22 to 28 years). Participants' demographic characteristics are presented in (S1 Table). Analysis of the transcribed data revealed six themes describing factors that affected the facilitation of students' learning about PPM. The themes include (1) *student motivation*, (2) *facilitators' attributes*, (3) *collaboration between academics and clinicians*, (4) *nurses' limited autonomy for decision-making regarding PPM practices*, (5) *shortage of human and material resources*, and (6) *educational qualification*.

## Student motivation

The findings revealed that nursing students' motivation was one of the factors influencing their competency acquisition for PPM. The study participants reported that lack of sufficient preparation for nursing students resulted in a lack of interest in their learning, and one nurse educator went on to say that *"the students seem to lack good guidance or good mobilization to be interested in pain management."* (NE1)

A couple of nurse preceptors pointed out that students should be responsible as part of their learning process, stating that:

> *"In this process, the role of the nurse is to help the student and there is also the role of the student: it requires students who are eager to learn about those competencies, and to check on their learning objectives. It is essential for the student to show me how to help her/him to achieve her/his learning objectives."* (P1)

> *". . .another thing is the students' willingness to learn. There are students. . . you understand that they are not all the same. One student may be inquisitive, eager to learn new things, asking trigger questions of how s/he would manage pain of the paediatric patient, and would say "tell me how it works?; what are we going to do-show me . . .."*

> *Such a student leaves the learning environment with gained knowledge regarding PPM."* (P2)

Students agreed that they should take the initiative to approach nurses in clinical settings to seek guidance for PPM practices. In this regard one of them said:

*"as students we have to go to the nurse and show our interest to learn; I can ask: I have seen that child is having pain, what can we do to help him, to alleviate his pain?. If we do not ask, the nurse will not come to us to teach how we will utilize pain management. We should show our commitment, meaning self-motivation."* (S1)

Participants further reported behaviours of some nursing students related to their perceived lack of motivation to learn during their clinical placement. One of the preceptors commented that some nursing students would dodge clinical practice when they could not see the clinical instructor (from the academic institution) around. S/he mentioned that:

*"I have seen some individual nursing students, instead of observing the nurse performing her/ his tasks to learn from them, or finding something else to do related to their objectives, leaving the ward. . . It was as if they did not know why they came (students) here into clinical placement. . ."* (P3)

## Facilitators' attributes

In this study, participants perceived that the attributes of facilitators i.e., nurse educators and preceptors, could influence the effective facilitation of students to learn PPM. It was mentioned that facilitators with personal qualities such as kindness and empathy create a supportive environment during the care of the paediatric patient and for the students so that students learn from their example. One of the preceptors said:

*". . .that is why I think that my students, the students I teach should also have that quality of. . . they should not see a person feeling pain, especially a child and let it pass without doing the right thing; I (facilitator) have to work toward that."* (P2)

A nurse educator reiterated this by maintaining that personality traits of facilitators contribute in one way or another to how s/he would facilitate students to acquire knowledge and skills. S/he said that:

*"Because I am humane, who I am and the effect of external factors such as what I have experienced in my life, what I have seen, what I have studied,. . .generally are a part of what helps me to support others, including the students I teach."* (NE2)

*"I can add that qualities such as kindness of the educator and of the preceptor reinforce the interpersonal relationship with our students and contribute to a conducive environment for their learning."*(NE3)

However, a nursing student indicated that this was not always the case as s/he mentioned about the nurse s/he worked within paediatric ward of one clinical setting:

*". . .the nurse did not listen to me or did not want to give some credit to what I was saying for the child whom I found had pain. I think it goes with her/his personal character."* (S2)

This was echoed by a preceptor who reported that some individual nurse preceptors may be unwilling to teach nursing students, stating that:

*"Although not many, some of nurses feel that the time spent helping students learn is wasted and prefer to focus on their routine tasks of providing patient care."* (P2)

## Collaboration between academicians and clinicians

Another factor that hampered the facilitation of students' acquisition of competency for PPM was a lack of strong collaboration between academics and clinical nurse preceptors. Seemingly, often the matter of students' guidance for competency acquisition is dealt with by each part without consultation, resulting in inconsistency in the way it is provided to students. For instance, it was indicated that clinical supervisors from nursing schools would go into the clinical setting, practice with the students, and would leave the clinical setting without asking for any feedback from clinical nurse preceptors about how they are guiding nursing students to acquire the skills.

*"they come (clinical supervisors/instructors), they practice, sign for the students and they go; there is no strong collaboration with us to help students learn."* (P4)

A nurse educator added that:

*"when I go to clinical settings I think about what I can do to improve [students' development] but sometimes nurses there do not have the same kind of thinking, which is a big challenge."* (NE3)

In another instance, the issue of a mismatch between what is taught in theory regarding PPM at nursing schools and pain control practices in the clinical area was raised by participants, indicating a poor match. Nursing students perceived that knowledge sharing was not done enough between academics and clinicians. This was supported by a couple of students who said that:

*"The problem is that we learn PPM here at school in theory, but when we go in clinical placement the practice is not sufficient; we don't learn the practice related to the theory learned."* (S2),

*". . . it is not easy for us students integrating the two situations to be able to practice, mostly because the way we learned the theory of pain management according to its severity is different from what we find at the clinical sites."* (S3)

## Nurses' limited autonomy for decision-making regarding PPM practices

The findings have indicated that nurses in the clinical setting lacked the autonomy for decision-making regarding paediatric pain control practices. Participants still perceived themselves as physicians' assistants with limited authority to choose paediatric pain relief practices. Participants considered that having to seek approval of physicians for any PPM interventions was an impediment to PPM competency guidance to students as their decision-making skills would be limited. It was summarised by a couple of nurse preceptors who expressed this concern:

*". . .in nursing, there is a really big challenge, depending on hospital policy. We are not allowed to decide on pain treatment, or to prescribe drugs without an order from the physician, so nursing students do not learn that much. . ."* (P3)

*"If the students end up knowing that they have the potential to do it and they do not have the right to make any decisions, and yet they see children suffering, it is discouraging. Essentially, it is essential to review the scope and to utilize the scope as soon as possible."* (P4)

However, another nurse voiced that they work collaboratively with medical doctors while dealing with pain in hospitalized children and there was no restriction to suggest a

change in managing the pain when the follow-up was well done and the assessment indicates that.

> "... it is a matter of evidence, when I correctly do the assessment and find that the child needs pain relief and the medical doctor is not nearby, I can give a painkiller to the child, notify the medical doctor and document it." (P1)

## Shortage of human and material resources

This theme describes how the shortage of resources i.e., shortage of human resources and the unavailability of materials affect the effective facilitation of students to learn PPM. The issue of imbalanced student to facilitator (clinical nurse preceptors) ratios due to a large number of nursing students and a limited number of supervisors (educators) from the academic institutions as well as the shortage of nursing staff in the clinical area was raised by many participants as a factor that impedes students learning about PPM. A preceptor stated that:

> "you can have many students in the department-they are coming from different schools and you are not able to spend the same time with each of them—it is also another challenge." (P5)

A couple of student echoed these thoughts:

> "When we reach the hospital, we find that we are a large number of students. You understand that it is hard for one supervisor to facilitate all the students s/he has on the clinical site performing certain practices regarding pain management, bearing in mind that the clinical supervisor attends the supervision only three days a week at the maximum." (S4)

> "... my colleague mentioned limited clinical supervisors; if one clinical supervisor has to supervise about six students from different years of the nursing education program, it is difficult to help all the students to achieve their respective clinical objectives including PPM." (S5)

Similarly, the shortage of nurses in the clinical area was pointed out as preceptors are overloaded by a large number of patients and do not have sufficient time to facilitate students' learning. A nurse preceptor reported that:

> "I have a lot of work with my patients that I am struggling to complete for that day. I won't have time concentrate on teaching the students." (P4)

The participants further expressed that facilitation of students' PPM competency acquisition was hindered by the shortage of specialised paediatric nurses among educators as well as clinical nurses, as it was mentioned by a nurse educator and a preceptor that:

> "My colleague who are educators really have challenges depending on their background; they do not have specialized knowledge of paediatric pain management. When you are teaching and you have to teach this kind of specific course, sometimes you don't put much effort into it as you are not a specialist." (NE4)

> "Limited staff resources are still challenging because there are not enough trained nurses who are specialized in managing children." (P6)

On the other hand, the unavailability of materials at nursing schools and in the clinical area affected the facilitation of PPM competency acquisition by students. This was highlighted by a couple of nursing student who stated that:

*". . .even if the teacher has given you a big package of knowledge sometimes you can miss the materials that you would use to alleviate that pain in practice."* (S6)

*"A nursing procedure or intervention may not be performed due to lack of appropriate resources; in this case we are not exposed to utilize that procedure or intervention and so as students our learning is affected."* (S7)

This was echoed by a nurse educator and a nurse preceptor who stated:

*"When I am teaching, I expect to teach the theory and the practical session in skills lab, but I ask myself where I am going to get the resources–such as manikins and other materials such as pain scales, protocols, etc. that will help me to teach pain management in children. It becomes tricky."* (NE5)

*"We have challenges for non-pharmacologic interventions such as distraction, toys and other materials that are not available in clinical areas."* (P5)

### Facilitators' educational qualifications

Findings from this study revealed that educational qualification among facilitators was another factor impacting the way students were guided to acquire nursing competencies. Participants mentioned that nurses who had a lower education qualification than the level of students in a nursing education program seemed to lack the confidence to assist those students. This was mainly reported by nurse educators and preceptors as one of them summarised it by stating that:

*"Nurses with a lower level of education have low self-esteem when it comes to facilitating students who are pursuing a higher nursing education program This is also another challenge."* (P3)

And another preceptor reiterated that:

*"The problem is low educational qualifications of some nurses who are solicited by students with a higher level of education., Sometimes you find that some do not have the capacity to provide guidance or to help the students to learn what they are supposed to be learning."* (P4)

It was further noted by a nurse educator that less qualified individual nurses and educators have a limited capacity to transfer knowledge and skills to students. S/he said that:

*"It is a concern because, among nurses, I have seen some who do not master pain assessment tools. They are not used to following up on children with pain to assess them to know the level of pain they are experiencing, and consequently which pain management interventions to perform for the children."* (NE6)

### Discussion

This study sought to explore factors influencing the facilitation of undergraduate nursing students in learning PPM as perceived by nurse educators, clinical nurses who are preceptors, and nursing students in Rwanda.

The data indicate that nursing students' learning of PPM was influenced by their motivation which can be aided by the appropriate orientation of the facilitators or the students'

intrinsic interest. Similarly, Mwale and Kalawa [35] reported that student motivation was an outstanding factor affecting clinical skill acquisition in nursing students. Bengtsson and Ohlsson [36] stated that extrinsic factors such as attitudes of educators and peer students have an important effect on students' motivation to learn new skills, which is congruent with the findings of the current study. The attributes of facilitators were highlighted in this study as having an influence on PPM competency acquisition by students. When nurses (educators and preceptors) act with kindness toward patients and students, the latter learn by example as the facilitators become role models. This was supported by previous literature that described facilitators as "agents of knowledge transmission, and role models for the next generation of professionals" [37, p.1941]. However, a recent study conducted in Rwanda [16] revealed that nursing students during their clinical placements witnessed inappropriate attitudes regarding PPM translating into ineffective role modelling by individual clinical nurses, aligning with the findings of the study conducted in Malawi [35], which found that poor role modelling affected the acquisition of competencies by nursing students. In the authors' understanding, supportive facilitators reinforce the interpersonal relationship with students which contributes to a more conducive learning environment. As stated by [38] "personality traits can help in the aligning of students and clinical instructors, as well as can cause rifts in the teaching relationship"(p.424). The traits of educators and preceptors were reported in other research [15,35] to be central to pain education.

From this study, there are indications that collaboration between nursing faculty and clinical nurses needs to be boosted as an ineffective collaboration between academics and clinicians may result in discrepancies between what is taught in the classroom and the practices in the clinical settings, thus hampering skills acquisition by students. The study findings suggest that knowledge sharing and collaborative practice among educators and clinical nurses is imperative to facilitate competency development among themselves and for student learning facilitation. Collaboration may minimize the discrepancy in performing nursing procedures between educators and clinical nurses. This is in agreement with findings from a study with clinical nursing instructors in Taiwan [39], highlighting that building a positive and cooperative relationship with clinical nurses was beneficial for clinical teaching and for creating a good working environment for the clinical instructor who is unfamiliar with the unit. Concerning pain education, previous literature has shown that the synergy between academics and clinicians influences the outcomes of students' learning activities [40].

The data indicated that a restrictive nursing scope of practice with regard to decision-making for PPM relief interventions was deemed to affect the acquisition of knowledge and skills by nursing students. Seemingly, nurses at some clinical settings felt they did not have professional autonomy to decide on paediatric pain relief practices, which is similar to what was reported in a previous study in which nurses voiced an issue with lacking the authority for prescribing analgesia to suffering children and were not working collaboratively in decision-making for patient pain management plans [41]. In the cases where individual clinical nurses do not have authority to decide on pain control interventions, there might be a lack of confidence from the patients or the caregivers in their abilities to deal with reported pain [41,42], which can impact the ability of nursing students to learn decision-making skills related to PPM in those settings as they miss the opportunity to do so. Notwithstanding, such concern was not raised by nurses working at teaching hospitals, probably because they may have a better understanding of their role in pain relief practices and see themselves more as collaborators in decision-making than doctors' order implementors. This indicates how the culture in certain practice environments can be limiting to nurses' abilities to engage more fully in PPM. Hence, it is important to handling high-level structural and leadership issues to settle such a situation. Rwanda Law Reform Commission [43] established healthcare professionals who can prescribe

narcotic drugs for analgesia purposes as required. The understanding of provisions from the Ministry of Health and health professional regulatory bodies could facilitate the implementation of this law within one's scope of practice while promoting interprofessional collaborative practices. Younas et al. [44] maintained that regulatory bodies ought to play a critical role and take a step forward toward the development of nursing as a discipline. Furthermore, an interprofessional education approach can be used to optimize shared decision-making toward PPM practices among healthcare professionals.

Another significant finding was the shortage of resources which was perceived as a hindrance for nursing students learning PPM. Similarly, previous research studies [41,45] revealed that a lack of basic resources prevented nurses from effectively performing their duties including PPM. Inadequate nursing staffing, as well as lack of essential materials such as paediatric pain assessment tools, can lead to improvising or at the worst to omitting a nursing procedure altogether. Kholowa et al. [46] maintained that lack of demonstration and practicing under supervision due to a shortage of paediatric pain assessment scales affected the acquisition of knowledge and attitudes regarding PPM. The authors in this study agree with previous researchers [47] who stated that when students are not able to learn a new skill or when they miss procedure steps it may result in they not being competent and being unsafe to practice.

The findings from this study demonstrate that the educational qualification of facilitators was of concern for nurse educators and preceptors. Participants expressed that less qualified individual nurses and educators had difficulties facilitating the students practicing what they have learned in theory about pain management because they have limited knowledge and skills. This finding supports earlier research evidence [48] confirming that facilitators with minimal pain education may transfer outdated knowledge and inappropriate attitudes to the next generation of clinicians. This finding also revealed that clinical nurses who had a lower education qualification than the education program pursued by the students seemed to lack self-confidence in assisting those students. The history of nursing education in Rwanda indicates that the training of nursing cadres at the tertiary education level started less than three decades ago, with an advanced diploma (A1) nursing program in 1996 and a bachelor's degree program in 2006. Before that, only enrolled nurses (A2) were trained in Rwanda until 2007 [12]. Initiatives to scale up the level of A2 nurses to A1 nurses were undertaken, although some health facilities are still staffed by A2 nurses working under the supervision of A1 nurses [49]. Within this particular context, a nursing student pursuing a bachelor of nursing educational program can likely be attended to by an A1 nurse. Unless the A1 nurse uses sound nursing practice experience enabling him/her to facilitate students' learning, s/he will teach the student the limited knowledge and skills s/he has. In the current study, some of the A1 nurses considered themselves as not qualified to teach students who might have a higher level of education, which is congruent with previous literature [46,50]. Therefore, while efforts to increase human resources are being strengthened, it is also important to enhance the self-efficacy of those who are already in service. Continuing education of clinical nurses can keep them updated on the current standard of nurse practices given that health care is dynamic and enhance their capacity to facilitate students' learning clinical skills.

## Conclusion

The present study reports factors that influence the facilitation of nursing students to learn PPM as perceived by nurse educators, preceptors, and students themselves. The identified factors include students' self-motivation, facilitators' attributes, the collaboration between nurse academics and nurse clinicians, restrictive nursing scope of practice regarding PPM, shortage

of resources, and the educational qualifications of facilitators. Knowing these factors is essential as it provides an opportunity to design targeted interventions aimed to enhance the capacity of nurse educators and clinical nurses involved in teaching nursing students. Such interventions should focus on giving students a proper orientation to increase their motivation for learning, and promoting interprofessional collaboration and shared decision-making in PPM. Furthermore, empowering and supporting nurses in both academic and clinical settings with required resources, and conducting continuing education should be considered to optimize paediatric pain relief practices by promoting a sound pedagogical approach to facilitate PPM skills acquisition by nursing students.

## Strengths and limitation

One of the strengths of this study is the use of different methods to collect data from the key informants to obtain information concerning factors affecting the facilitation of students in learning PPM. Individual interviews with nurse educators and preceptors and FGDs with nursing students, as the main stakeholders involved in the teaching-learning experience, were conducted to gain a comprehensive understanding of the phenomenon under study. Another strength is that these key informants were from different settings including public and private educational institutions as well as health facilities such as rural district hospitals, referral hospitals, and university teaching hospitals. However, this study could be limited by the fact that the identified factors affecting the facilitation of students to learn PPM represent personal views of the study participants in the context of Rwanda, therefore, further work is required for a possible generalization of the finding to a different context in the region and internationally.

## Supporting information

**S1 File. Individual interview guide in English and Kinyarwanda.**
(DOCX)

**S2 File. Interview guide for FGDs in English and Kinyarwada.**
(DOCX)

**S1 Table. Demographic characteristics.**
(DOCX)

## Acknowledgments

The authors are grateful to the educators, preceptors, and students in nursing for their voluntary participation in the study. Also, our gratitude goes to Mrs Renee Pyburn of SLO South County Behavioural Health Services (California, USA) who read and provided helpful suggestions for editing the manuscript.

## Author Contributions

**Conceptualization:** Philomene Uwimana, Donatilla Mukamana, Yolanda Babenko-Mould, Oluyinka Adejumo.

**Data curation:** Philomene Uwimana, Donatilla Mukamana, Oluyinka Adejumo.

**Formal analysis:** Philomene Uwimana, Donatilla Mukamana, Yolanda Babenko-Mould, Oluyinka Adejumo.

**Funding acquisition:** Philomene Uwimana.

**Investigation:** Philomene Uwimana.

**Methodology:** Philomene Uwimana, Donatilla Mukamana, Yolanda Babenko-Mould, Oluyinka Adejumo.

**Resources:** Philomene Uwimana.

**Software:** Yolanda Babenko-Mould.

**Supervision:** Donatilla Mukamana, Yolanda Babenko-Mould, Oluyinka Adejumo.

**Validation:** Philomene Uwimana, Donatilla Mukamana, Yolanda Babenko-Mould, Oluyinka Adejumo.

**Writing – original draft:** Philomene Uwimana, Donatilla Mukamana, Yolanda Babenko-Mould, Oluyinka Adejumo.

**Writing – review & editing:** Philomene Uwimana, Donatilla Mukamana, Yolanda Babenko-Mould, Oluyinka Adejumo.

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
