## [Decision Letter · Decision Letter 0]

25 Oct 2021

PONE-D-21-20434Exploring factors affecting the facilitation of nursing students to learn paediatric pain management in Rwanda: A descriptive qualitative study.PLOS ONE

Dear Dr. Uwimana,

Thank you for submitting your manuscript to PLOS ONE. After careful consideration, we feel that it has merit but does not fully meet PLOS ONE’s publication criteria as it currently stands. Therefore, we invite you to submit a revised version of the manuscript that addresses the points raised during the review process.

We look forward to receiving your revised manuscript.

Kind regards,

Khatijah Lim Abdullah, DClinP, MSc., BSc

Academic Editor

PLOS ONE

Reviewers' comments:

Reviewer's Responses to Questions

5. Review Comments to the Author

Reviewer #1: PONE-D-21-20434

Overall a clear and nicely presented manuscript on an important subject however clarifications must be made in regard to methodology!

Keywords: Should they be presented in alphabetical order?

INTRODUCTION

Line 39- 41: A reference would support this important statement which is a rational for the entire study.

Line 68 (+ abstract): could “the effective” be omitted in the aim? …”explored factors affecting the effective facilitation of nursing students for learning about PPM as perceived by academics, clinical nurse preceptors, and nursing students in Rwanda.”

METHODS

Line 76-77: Is this sentence “The study authors utilised this approach to explore the factors influencing the facilitation of nursing students’ learning about PPM” needed or just a way of rephrasing the aim?

The choice of qualitative approach was unclear to me when I first read the manuscript. Reference 15; Neergaard (2009) Qualitative description-the poor cousin of health research? as well as Qualitative Descriptive (QD) as an approach was unknown to me and I applogize if I that might interfere with some of my comments. However I have now read the reference and tried my best to understand the manuscript in relation to this. Further down in the methods authors claim that they have used a thematic analysis (TA). Neergaard et al. describe the possibility to within QD use the steps of content analysis within QD. Is it based on this information the authors of this manuscript have decided to include also TA? If so, the manuscript would benefit from a description of QD and a rational for why these two approach was chosen. Other parts of the method section should also benefit from adjustments in relation to QD. For example; have the authors really performed in-depth interviews and if so why when Neergaard promote semi-structured interviews. Is it really narratives that has been collected or rather experiences?

Line 79: Authors state that the sites were chosen purposively and by the description it seems as if authors were choosing purposively to reach a variation in included sites. This could be clarified.

Line 104: The statement that data was saturated after 6+8 informants is quite specific and the sentence would be strengthened by some further explanation on the matter. In what regard was the data saturated? Who decided? Is it in line with Neergaard et al./ QD to reach saturation?

Line 124: It is unclear to me what the reference is supporting? Total number of participants? The sentence is not telling us anything about focus group size. Furthermore somewhere in the method and/or discussion the choice of using both focus group and individual interviews should be described or reflected on.

Line 130: Can an interview be both semi structured and in-depth as previously stated? Could the authors clarify the content of the interview guide. It is stated in the manuscript, If I have understood it correctly, that this was built on findings from previous studies. In what way or to what extent??

Line 139-140; Could the information be more specific than generally 30-60 minutes?

Line 149-151: With whom were they provided this opportunity, when and why?

Line 154: The use of codes and pseudonyms can only be considered anonymous if there is no code-key or anything otherwise consider changing to “confidentiality”.

Line 175: It is unclear to me what authors mean by “the codes were combined to form larger meaningful segments using the deductive analytic technique that enabled the conceptualization of themes”. Was a deductive analysis really performed and if so based on what theory or model? It does not correspond to other statements in the method section.

RESULTS

Line: 218: Is this part of the quote adequate for the theme? ”Such a student leaves the learning environment with gained knowledge. So, students have to be well focused when they come into clinical placement with daily objectives that they have to achieve.”? Or could this last part be removed?

Line 235: Is it clear who is the facilitator in this case?

Line 288: Could this Theme be rephrased? The content is interesting but the title “Nurses' restrictive scope of practice…” is unclear to me.

Line 290-292: Is the sentence “To some extent nurses are still perceived as physicians’ assistants with limited authority to choose paediatric pain relief practices.” derived from data or is it a statement from the authors? Possibly rephrase the sentence for clarification. (If from the authors remove)

Line 306: Could this Theme be rephrased to be more specific The recourses discussed is human and material resources.

DISCUSSION

Wordy but overall relevant and nice discussion

Line 400-4001: Should the following be included in the results in order to be discussed? “As stated by [31] “personality traits can help in the aligning of students” and clinical instructors, as well as can cause rifts in the teaching relationship”(p.424)

STRENGTHS AND LIMITATIONS

Line 496-499: I would not consider it a limitation for the study as such that it “represent only personal views of the study participants” as this was the purpose of the study.

References:

Most references are at least 3 years old. Would it be possible to supplement with more resent research findings?

Reviewer #2: A clear, articulate introduction and background which states and validates the research study’s importance in respect to aspects related to global and local pediatric safety and patient advocacy. The materials and methods sections were assessed as follows:

• Study Design and Setting was clear and comprehensive

• Selection of Participants was clear and appropriate

• Data Collection Procedures was clear and thorough

• Data Analysis was clear and trustworthy

• Validity and rigor were clear and reliable

• Results were interesting and unique in the context of Rwanda

• The discussion was insightful and provides an opportunity for self-reflection pertaining to clinical practices in pediatric nursing.

• Strengths and limitations were appropriate.

Well done

---

## [Author Response · Author response to Decision Letter 0]

10 Dec 2021

Comments given by the reviewer in the methods section about use of semi-structured interviews, about data saturation and thematic analysis in qualitative descriptive, and the coding process were addressed with supporting new references. The file "Response to reviewers" contains all the information related on how it was done and the pages and lines are provided for reference in the manuscript with track changes.

The comments from the editor included to follow the journal format for the manuscript, to provide the right number of grant that supported the study, and specify if there are ethical or legal restrictions on sharing a de-identified data set or if there are no restrictions, to upload the minimal anonymized data set necessary to replicate our study finding. These comments were also addressed, the manuscript is in the right format, the number of the grant is provided in the "Funding Information" section, and Supporting Information files are uploaded.

---

## [Decision Letter · Decision Letter 1]

24 Jan 2022

Exploring factors affecting the facilitation of nursing students to learn paediatric pain management in Rwanda: A descriptive qualitative study.

PONE-D-21-20434R1

Dear Dr. Uwimana,

We’re pleased to inform you that your manuscript has been judged scientifically suitable for publication and will be formally accepted for publication once it meets all outstanding technical requirements.

Kind regards,

Khatijah Lim Abdullah, DClinP, MSc., BSc

Academic Editor

PLOS ONE

Additional Editor Comments (optional):

Reviewers' comments:

Reviewer's Responses to Questions

**Comments to the Author**

6. Review Comments to the Author

Reviewer #1: The authors have made a rigour revision and improved the manuscript. Please review for typographical and grammatical errors

---

## [Editor Report · Acceptance letter]

8 Feb 2022

PONE-D-21-20434R1 

Exploring factors affecting the facilitation of nursing students to learn paediatric pain management in Rwanda: A descriptive qualitative study. 

Dear Dr. Uwimana:

I'm pleased to inform you that your manuscript has been deemed suitable for publication in PLOS ONE. Congratulations! Your manuscript is now with our production department. 

Kind regards, 

on behalf of

Dr. Khatijah Lim Abdullah 

Academic Editor

PLOS ONE